# Honokiol-Rich *Magnolia officinalis* Bark Extract Attenuates Trauma-Induced Neuropathic Pain

**DOI:** 10.3390/antiox12081518

**Published:** 2023-07-28

**Authors:** Vittoria Borgonetti, Nicoletta Galeotti

**Affiliations:** Department of Neuroscience, Psychology, Drug Research and Child Health (NEUROFARBA), Section of Pharmacology and Toxicology, University of Florence, Viale G. Pieraccini 6, 50139 Florence, Italy; vittoria.borgonetti@unifi.it

**Keywords:** neuropathic pain, honokiol, *Magnolia officinalis*, antioxidant, cannabinoid receptor, neuroinflammation

## Abstract

Neuropathic pain (NP) affects about 8% of the general population. Current analgesic therapies have limited efficacy, making NP one of the most difficult to treat pain conditions. Evidence indicates that excessive oxidative stress can contribute to the onset of chronic NP and several natural antioxidant compounds have shown promising efficacy in NP models. Thus, this study aimed to investigate the pain-relieving activity of honokiol (HNK)-rich standardized extract of *Magnolia officinalis* Rehder & E. Wilson bark (MOE), well known for its antioxidant and anti-inflammatory properties, in the spared nerve injury (SNI) model. The molecular mechanisms and efficacy toward neuroinflammation were investigated in spinal cord samples from SNI mice and LPS-stimulated BV2 microglia cells. MOE and HNK showed antioxidant activity. MOE (30 mg/kg p.o.) produced an antiallodynic effect in SNI mice in the absence of locomotor impairment, reduced spinal p-p38, p-JNK1, iNOS, p-p65, IL-1ß, and Nrf2 overexpression, increased IL-10 and MBP levels and attenuated the Notch signaling pathway by reducing Jagged1 and NEXT. These effects were prevented by the CB1 antagonist AM251. HNK reduced the proinflammatory response of LPS-stimulated BV2 and reduced Jagged1 overexpression. MOE and HNK, by modulating oxidative and proinflammatory responses, might represent interesting candidates for NP management.

## 1. Introduction

Chronic pain is a type of pain that persists for more than 3 months, as defined by the International Association for the Study of Pain (IASP), that can be classified by its origin as cancer pain, post-surgical pain, neuropathic pain, oncoplastic pain, visceral pain, musculoskeletal pain, etc. While chronic pain of inflammatory origin has many therapeutic options, currently available treatments for neuropathic pain (NP) are limited and usually accompanied by severe side effects, making their use controversial [1]. The characteristics of NP are hyperalgesia and allodynia and several comorbidities such as anxiety and depression [2,3]. The complexity of the symptomatology makes NP the most difficult to treat chronic pain condition and novel and effective strategies to prevent this condition are an urgent medical need.

Even though the mechanisms responsible for the induction and maintenance of NP are not fully elucidated, evidence indicates excessive oxidative stress and oxidative species formation can contribute to the onset of chronic NP. In this condition, there is an increase in the endogenous reactive oxygen species (ROS) production that overwhelms a cell’s antioxidant capability leading to cellular damage [4]. An important ROS target is polyunsaturated fatty acids, which are extremely sensitive to oxidation. Thus, the high sensibility of neurons to oxidative damage originates from the high concentrations of axonal mitochondria, high cell membrane concentrations of phospholipids, and weak antioxidant defenses [5], which makes oxidative stress a relevant target to NP management [6].

Antioxidants obtained from natural sources include low-molecular-weight compounds, such as vitamins, plant-derived polyphenols, glutathione, and carotenoids. In recent years, the use of plant-derived preparations containing high levels of polyphenols for the management of pain has increased thanks to their lower side effects compared to synthetic compounds [7]. Several natural antioxidant compounds have shown promising efficacy in different models of NP, such as gingerols and shogaols [8,9], curcumin [10], resveratrol [11], and others [12]. Numerous studies are currently focused on the characterization and application of natural agents in various diseases for the reduction in and/or elimination of free radicals [13]. Furthermore, natural antioxidants have contributed to developing new therapeutic interventions for the treatment of chronic NP [14].

The bark of *Magnolia officinalis* Rehder & E. Wilson, also known as “Houpo” in Chinese, has been used for thousands of years in traditional Eastern medicine, boasting more than 2500 years of history [15]. In Eastern culture, the uses of *Magnolia officinalis* extracts are as varied as treating depression, anxiety, nervous disorders, gastrointestinal disorders, asthma, and stroke, and for relieving migraine, muscle pain, and fever [15]. The active component of *Magnolia officinalis* extracts (MOE) are mainly polyphenolic neolignans, such as magnolol and honokiol [16,17]. These compounds are known for their wide range of pharmacological effects that are largely attributed to their antioxidant activity [18]. In particular, magnolol and honokiol inhibited oxygen consumption and malondialdehyde production by lipid peroxidation with a 1000 times higher efficacy than α-tocopherol [19].

The imbalance between the oxidative and antioxidant mechanisms of the body leads to oxidative stress and promotes inflammation. Evidence indicates the involvement of proinflammatory mediators and microglia-mediated neuroinflammation as processes underlying nociceptive responses [20,21], including conditions of NP [2]. Thus, treatments endowed with anti-inflammatory and antioxidant activities could represent a promising therapeutic option. In addition to the antioxidant activity, honokiol (HNK) displayed anti-inflammatory action in primary cultures of microglia and astrocytes stimulated by LPS, as demonstrated by the inhibition of expression of pro-inflammatory mediators, such as IL-6, IL-1β, iNOS, and TNF-α [22] and showed antinociceptive activity in inflammatory pain models [23]. Furthermore, a recent study showed the efficacy of magnolin, a neolignan obtained from the flower buds of *Magnolia denudata*, in attenuating paclitaxel-induced cold allodynia [24], In the effort to identify new potential therapies for NP, the purpose of our research was to evaluate the analgesic activity of an HNK-rich standardized extract of *Magnolia officinalis* Rehder & E. Wilson bark (MOE) in an animal model of neuropathic pain. The underlying molecular mechanisms of MOE and HNK were investigated in vitro. It has been observed that MOE and its main constituents magnolol and HNK can activate cannabinoid receptors. Specifically, magnolol behaved as a partial agonist for CB2 receptors, while HNK showed full agonistic activity at CB1 receptors [25]. Since the activation of the cannabinoid system promotes potent antioxidant and anti-inflammatory activity [26], we also assess whether the analgesic effect might depend on the modulation of this receptor class.

## 2. Materials and Methods

### 2.1. Drug Administration

*Magnolia officinalis* Rehder & E.H. Wilson bark extract (MOE, extraction solvent: ethanol 96% *v*/*v*, standardized to contain 40% honokiol, Naturex Inc., South Hackensack, NJ, USA) was used. For behavioral tests, MOE was dissolved in saline (0.9% NaCl) and administered in a volume of 10 mL/kg by gavage (p.o.) 45 min before the experimental procedure at the dose of 30 mg/kg, except for the dose–response curve where doses of MOE ranged from 1 to 60 mg/kg. For in vitro studies MOE was dissolved in Dulbecco’s modified Eagle’s medium (DMEM) (Sigma Aldrich, Milan, Italy) to reach the final concentrations of 0.1, 1, and 10 μg/mL, according to cell viability experiments for cell culture studies. AM251 (Tocris, Bristol, UK), a known CB1 antagonist, was dissolved in dimethyl sulfoxide/Tween 80/0.9% saline (1:1:18) and administered (0.5 and 3 mg/kg i.p.) 30 min before behavioral tests. HNK (0.1–10 μg/mL; Sigma Aldrich, Milan, Italy) and DAPT (3 µM; Sigma Aldrich, Milan, Italy) were dissolved in 0.5% DMSO. Pregabalin (30 mg/kg i.p.; Sigma Aldrich, Milan, Italy), used as an antiallodynic reference drug, was dissolved in saline solution and administered 3 h before testing.

### 2.2. Animals

CD1 male mice (20–22 g) from the Envigo (Varese, Italy) were used. Mice were randomly assigned to standard cages, with four to five animals per cage. The cages were placed in the experimental room 24 h before behavioral testing for acclimatization. The animals were fed a standard laboratory diet and tap water ad libitum and kept at 23 °C with a 12 h light/dark cycle, light on at 07:00 h. All animal care and experimental protocols complied with international laws and policies (Directive 2010/63/EU of the European Parliament and of the Council of 22 September 2010 on the protection of animals used for scientific purposes; Guide for the Care and Use of Laboratory Animals, U.S. National Research Council, 2011). Animal studies are reported in compliance with the animal research: reporting of in vivo experiments (ARRIVE) guidelines [27,28]. All efforts were taken to minimize the number of animals used and their suffering. Mice were sacrificed by cervical dislocation for removal of the spinal cord for in vitro analysis. The number of animals per experiment was based on a power analysis and calculated by G power software 3.1.9.6 [29]. To determine the anti-nociceptive effect, each tested group comprised eight animals.

### 2.3. SNI Model 

As previously reported [30], mice were anesthetized with a mixture of 4% isoflurane in O_2_/N_2_O (30:70 *v*/*v*) and placed in a prone position. The right hind limb was slightly elevated, and a skin incision was made on the lateral surface of the thigh. The sciatic nerve was exposed and both tibial and common peroneal nerves were ligated with microsurgical forceps (5.0 silk, Ethicon; Johnson & Johnson Intl, Brussels, Belgium) and transacted together. Uninjured sural extensions are used to measure the mechanical allodynia and thermal hyperalgesia associated with the model. The sham procedure consisted of the same surgery without ligation and transection of the nerves.

### 2.4. Nociceptive Tests

Behavioral nociceptive tests were performed before surgery, to establish a baseline for comparison with postsurgical values, and 7 days after surgery by a blinded operator.

#### 2.4.1. Hot Plate Test

The hot plate test involves the evaluation of thermal hyperalgesia using a circular metal surface (24 cm diameter) electrically heated to a temperature of about 52.5 °C. The mice are placed on the hot plate surrounded by a transparent acrylic cage, and the response time of the animals to the hyperalgesic stimulus is measured. Response latency (measured in s) consists of a leap, licking, or shaking of the paw. The mouse is immediately removed from the plate when it exhibits any of these symptoms. The animals were tested one at a time and did not undergo a period of adaptation to the experimental system before testing [31].

#### 2.4.2. Von Frey Filaments

The Von Frey test was used to evaluate mechanical allodynia [32]. The tests were carried out both before the operations, using the data obtained as a reference, and afterward for comparison. Mechanical nociception was measured by Von Frey monofilaments. Mice were placed in single plexiglass chambers [8.5 × 3.4 × 3.4 (h) cm]. After a settling period of 1 h inside the chambers, the mechanical threshold was measured through a stimulus using Von Frey monofilaments with increasing degrees of strength (0.04, 0.07, 0.16, 0.4, 0.6, 1.0, 1.4, 2.0 g) on both legs, ipsilateral and contralateral. The response was defined by the withdrawal of the paw three times out of five stimuli performed. In the event of a negative response, the next higher-grade strand was applied, and the averages of the responses were finally calculated.

### 2.5. Evaluation of Locomotor Side Effects

#### 2.5.1. Rotarod Test

Rotarod was used to assess any impairment to the mouse motor coordination induced by the treatments. The test was performed before and 15, 30, 45, 60, 90, and 120 min after oral administration in naïve mice. The motor function was evaluated by counting the number of falls in 30 s, as previously described [33].

#### 2.5.2. Hole Board Test

The spontaneous mobility and the exploratory activity were recorded as counts in 5 min using the hole board test, as described [33]. The test was performed 45 min after oral administration in naïve mice. 

### 2.6. DPPH Radical Scavenging Assay

The capacity of MOE and HNK to scavenge 1,1-diphenyl-2-picryl-hydrazyl (DPPH) free radicals was measured by a spectrophotometric method. Crude extract and HNK were solved in methanol. One hundred microliters of extract, or HNK at different concentrations, were mixed with 100 μL of a DPPH methanolic solution (0.04 mg/mL). Methanol served as a blank and DPPH in methanol without the extract or HNK served as positive control. Ascorbic acid was used as a reference drug. After 30 min of reaction at room temperature, the absorbance was measured at 517 nm. Controls contained all the reaction reagents except the plant extract, honokiol, or reference substance. Background interferences from solvents were deducted from the activities of the corresponding extracts before calculating radical scavenging capacity as follows:Radical scavenging activity (%) = [(Abs_control_ − Abs_sample_)/Abs_control_] × 100

The antioxidant activity of honokiol was expressed as IC_50_, which is defined as the concentration (μg/mL) required to scavenge 50% of DPPH radicals. IC_50_ values were estimated by non-linear regression (GraphPad Prism version 10.0). A lower IC_50_ value indicates higher antioxidant activity. The results are given as a mean ± SEM of experiments done in triplicate.

### 2.7. Preparation of Tissue and Cell Lysates

Proteins extraction from tissues and cells was performed as previously reported [32]. Briefly, mice were sacrificed 7 days after SNI, and the lumbar spinal cord tissue was removed. Samples were homogenized in a lysis buffer containing 0.1% SDS (Sigma-Aldrich, St. Louis, MI, USA). The homogenate was centrifuged at 12,000× *g* for 30 min at 4 °C, and the pellet was discarded. Proteins from BV2 cells were extracted by radioimmunoprecipitation assay buffer (RIPA) buffer (Sigma-Aldrich), and the insoluble pellet was separated by centrifugation (12,000× *g* for 30 min, 4 °C). The total protein concentration in the supernatant was measured using the Bradford colorimetric method (Sigma-Aldrich, St. Louis, MO, USA). 

### 2.8. Western Blotting

Protein samples (20–40 µg of protein/lane) were separated by 10% SDS PAGE [34]. Proteins were then blotted onto nitrocellulose membranes (120 min at 100 V) using standard procedures. Membranes were blocked in PBS with 1% tween 20 (PBST) containing 5% non-fat dry milk for 120 min and incubated overnight at 4 °C with primary antibodies p-ERK1/2 (Thr202/Tyr204) (1:1000; Cell Signaling, Danvers, MA, USA), p-p38 (Thr180/Tyr182) (1:1000; Cell Signaling, Danvers, MA, USA), p-JNK1 (Thr183/Tyr185) (1:750; Cell Signaling, Danvers, MA, USA), p-p65 (Ser536) (1:1000; Santa Cruz Biotechnologies, Dallas, TX, USA), iNOS (1:500; Santa Cruz Biotechnologies, Dallas, Texas, USA), anti-IL-1β (1:1000; Bioss Antibodies, Woburn, MA, USA), IL-10 (1:500; Santa Cruz Biotechnologies, Dallas, TX, USA); Jagged1 (1:500; Santa Cruz Biotechnologies, Dallas, TX, USA); NEXT (1:500; Santa Cruz Biotechnologies, Dallas, TX, USA); MBP (1:1000; Santa Cruz Biotechnologies, Dallas, TX, USA); Nrf2 (1:1000; Santa Cruz Biotechnologies, Dallas, TX, USA). The blots were rinsed three times with PBST and incubated for 2 h at rt with HRP-conjugated mouse anti-rabbit (1:3000; Santa Cruz Biotechnology, Dallas, TX, USA) and goat anti-mouse (1:5000, bs-0296G, Bioss Antibodies, Woburn, MA, USA) and then detected by a chemiluminescence detection system (Pierce, Milan, Italy). For each sample, the signal intensity was normalized to that of total protein stained by Ponceau S, and the acquired images were quantified using ImageJ software (Version 2.9.0).

### 2.9. BV2 Cell Culture 

A murine microglial line BV-2 (mouse, C57BL/6, brain, microglial cells, Tema Ricerca, Genova, Italy) was used for this study. The cells were thawed and placed in a 75 cm^2^ flask (Sarstedt, Milan, Italy) in a medium containing RPMI with the addition of 10% of heat-inactivated (56 °C, 30 min) fetal bovine serum (FBS, Gibco^®^, Milan, Italy) and 1% glutamine. Cells were grown at 37 °C and 5% CO_2_ with daily medium change [32]. Microglial cells were seeded in 6-well plates (3 × 10^5^ cells/well) until 70–80% confluence was achieved. The cells were then pretreated with MOE (0.1–10 µg mL^−1^), HNK (3 µg mL^−1^), or DAPT (SAHA, 3 µM) for 4 h and then stimulated with LPS, (250 ng mL^−1^) for 24 h. 

### 2.10. CCK-8 Test

Cell viability was performed using a Cell Counting Kit (CCK-8, Darmstadt, Germany) according to the manufacturer’s instructions. A total of 5.0 × 10^5^ cells/well were seeded into 96 multi-well plates and grown to confluence. The absorbance was measured at 450 nm using a microplate photometer HiPo MPP-96 (Biosan, Riga, Latvia). The treatments were performed in six replicates in three independent experiments, and the cell viability was calculated by normalizing the values to the control’s mean [35].

### 2.11. Statistical Analysis

Data and statistical analysis in this study comply with the recommendation on experimental design and analysis in pharmacology [36]. For in vitro analysis, data are expressed as the mean ± SEM of five experiments done in triplicate and assessed by one-way and two-way ANOVA followed by Tukey post hoc test. The behavioral data are presented as means ± SEM. Two-way analysis of variance (ANOVA) followed by Bonferroni post hoc was used for statistical analysis. For the locomotor activity, the unpaired sample *t*-test was performed. The software GraphPad Prism version 9.0 (GraphPad Software, San Diego, CA, USA) was used in all statistical analyses. For each test, a value of *p* < 0.05 was considered significant.

## 3. Results

### 3.1. Effect of MOE on Pain Threshold

#### 3.1.1. Antinociceptive Activity of MOE

The efficacy of MOE in attenuating the nociceptive behavior was evaluated after applying an acute thermal stimulus to naïve mice or in conditions of chronic pain (neuropathic mice). 

The analgesic activity of MOE was investigated in the hot plate test. The dose-response study showed that MOE produced a dose-related antinociceptive activity against an acute thermal stimulus. The doses of 1 and 10 mg/kg p.o. were ineffective. The dose of 20 mg/kg significantly increased the pain threshold, whereas the dose of 30 mg/kg reached the peak of the effect. No further increase was detected at higher doses (Figure 1A). Time-course experiments showed that the antinociceptive effect of MOE 30 mg/kg peaked 45 min after oral administration and then diminished, before disappearing at 90 min (Figure 1B). The increase in pain threshold produced by MOE after 45 min from the administration was compared to that produced by morphine (Figure 1C), used as an analgesic reference drug. 

#### 3.1.2. Antiallodynic Effect of MOE

In the SNI model, there is a similar microglia proliferation in the dorsal horn in both sexes. However, females use a microglia-independent pathway to mediate pain hypersensitivity [37]. In addition, SNI male mice, but not female, display long-lasting neuropathic pain behavior selectively related to a male-specific increase of cellular senescence markers in microglia [38]. The present study aims to investigate the contribution of MOE on modulating the microglia-mediated neuroinflammation and we performed experiments on male mice.

The antiallodynic effect of MOE in conditions of NP was evaluated in a mouse model of peripheral neuropathy, the SNI. Seven days after surgery, SNI mice showed marked mechanical hyperalgesia on the ipsilateral side in comparison with the contralateral uninjured side. Dose–response studies showed MOE at the doses of 20 and 30 mg/kg significantly decreased the mechanical allodynia (Figure 2A). Time-course analysis revealed that MOE showed a significant anti-allodynic effect 30 and 45 min after oral administration. This effect was largely reduced at 60 min and drastically disappeared after 90 min, with a time course similar to that observed for acute pain (Figure 2B). The intensity of the MOE effect was compared to that produced by pregabalin (30 mg/kg), a well-known neuropathic pain-relieving agent, used as a reference drug (Figure 2C). 

The role of the cannabinoid system in the mechanism of action of MOE was investigated by pretreating animals with the CB1 antagonist AM251, administered at the dose of 0.5 mg/kg that was devoid of any effect on pain threshold when administered alone (Figure 2D). The antiallodynic activity of MOE (30 mg/kg per os) was completely abolished when co-administered with the CB1 antagonist AM251 (Figure 2E). This finding suggests that the analgesic effect of MOE extract in neuropathic pain is related to a direct or indirect action on CB1.

### 3.2. Evaluation of Side Effects on Motility and Exploratory Ability of MOE

The time course on the locomotor effects of MOE 30 mg/kg was assessed by the number of falls from the rotating rod with the rotarod test. The number of falls was progressively reduced in the control group since animals learned how to balance on the rotating rod, thus minimizing the number of falls. MOE did not induce any motor impairment as the reduction in the number of falls from the pretest values, is comparable to that of control mice (Figure 3A). Figure 3B shows that administration of the extract at peak activity did not cause alteration of normal exploratory activity or spontaneous mobility in the animals. These results lead us to exclude the possible side effects of MOE on locomotor behavior and possible sedative effects.

### 3.3. Effect of MOE on the Phosphorylation of MAPK

To understand the mechanism of action of MOE, the expression of the phosphorylated form of mitogen-activated protein kinases (MAPK) in spinal cord samples was evaluated (Figure 4). Results showed the levels of p-ERK1 and p-ERK2 in spinal cord samples from the ipsilateral injured side (IPSI) were abundantly overexpressed than in the contralateral side (CONTRA). SNI mice treated with MOE at the dose of 30 mg/kg showed a non-significant trend toward a reduction of p-ERK1/2. SNI mice showed a significant increase in p-p38 in the ipsilateral side compared to the CONTRA. MOE administration induced a reduction in p-p38 levels, bringing them back to values comparable to the control. Finally, mice with neuropathy (SNI) exhibited a significant increase in p-JNK1 expression in the injured side of the spinal cord compared with the healthy contralateral side, which was drastically reduced by MOE treatment. 

To investigate the role of CB1 receptors in the anti-allodynic action of the extract, mice were pretreated with AM251. Co-administration of MOE and AM251 did not show a substantial difference from the administration of MOE alone against p-ERK1/2 and p-JNK1 whereas a complete reversal of the MOE effect was observed for p-p38 (Figure 4), thus suggesting an important role of CB1-mediated events in the p38-related mechanisms. 

### 3.4. Effect of MOE on Inflammatory Markers and Oxidative Stress in Spinal Cord Samples from SNI Mice

A substantial increase in inducible nitric oxide synthase (iNOS) expression was observed in SNI mice compared to control (Figure 4). MOE administration caused a strong reduction in iNOS levels, with values similar to those recorded for the control group. SNI mice had drastically increased expression of p-p65, a transcriptional factor able to promote the inflammatory response, on the ipsilateral side. MOE significantly reduced the activation of p-p65, lowering the value to the control group level. SNI procedure also increased the levels of IL-1β in the ipsilateral side that were reduced by MOE. All these effects on proinflammatory mediators were prevented by AM251 pretreatment (Figure 5). SNI mice also showed a reduction in the expression of IL10, an anti-inflammatory cytokine, that was increased by MOE treatment through an AM251-independent mechanism (Figure 5). 

### 3.5. Effect of HNK on LPS-Stimulated Microglia Cells

Neuroinflammation has been demonstrated to strongly participate in the development and progression of neuropathic pain symptoms, and microglia play an important role in this pathological context [39]. To better elucidate the mechanism of MOE activity and to identify the main constituents involved in its effect, we investigated the role played by honokiol (HNK) on microglia activation in BV2 cells. Exposure of microglia cells to LPS induced a reduction in cell viability that was dose-dependently reduced by pretreatment with HNK (0.1–10 µg/mL). The improvement in cell viability reached statistical significance at 1 µM and peaked at 3 µg/mL. No further increase in efficacy was detected at higher doses (Figure 6A). HNK (3 µg/mL) was also able to restore the normal cell morphology of cells. Exposure to LPS increased the cell surface area as well as the number of cells in the proinflammatory phenotype. HNK treatment reduced both cell surface area (Figure 6B) and the number of activated cells (Figure 6 C,D) in LPS-stimulated BV2. 

### 3.6. Evaluation of the Effect of MOE and HNK on the Notch Signaling Pathway

Numerous reports have shown that the Notch signaling pathway is closely related to microglial activation and differentiation and might play a role in central nervous system diseases [40]. Thus, we investigated the role of the Notch signaling pathway in the MOE activity on BV2 cells. Figure 7A showed the effect on cell viability of increasing doses of MOE to exclude the induction of toxicity. MOE (01–10 µg/mL) never reduced cell viability. Thus, in the following experiment, a dose of 10 µg/mL was used. The levels of Jagged1 were measured in LPS-stimulated BV2 cells that showed a robust increase in protein expression compared to the untreated cells. MOE reduced the Jagged1 expression to CTRL levels. The same results were obtained with HNK, administered at the dose present in the extract, indicating that it might represent the main constituent responsible for this effect. We compared the activity of MOE with DAPT, a γ-secretase inhibitor, largely used as a Notch signaling pathway inhibitor. MOE and HNK activity was exerted with an efficacy comparable to that showed by DAPT (3 µM) (Figure 7B). 

To further assess the role of Notch signaling in the attenuation of neuroinflammation by MOE, the effect of the extract on the expression of Jagged1 and Notch1 proteins was detected in the spinal cord of mice with SNI-induced neuropathy. Increased expression of Jagged1 (Figure 7C) and NEXT (Figure 7D) was observed in the ipsilateral side of the spinal cord of untreated SNI mice, suggesting an injury-induced increase in this pathway. MOE-treated animals showed lower protein levels, compared to the untreated group, for both Jagged1 and NEXT, indicating a dampening of the activity of the pathway under investigation. Co-administration of MOE and AM251 significantly reversed this effect (Figure 7C,D). 

### 3.7. Antioxidant Activity of MOE and HNK

The active doses of MOE and HNK on BV2 cells were tested for antioxidant activity in the DPPH test. MOE showed an antiradical scavenger activity at 10 µg/mL that was comparable to that produced by HNK at 3 µg/mL (Figure 8A). These results were consistent with the composition of the extract which contains 40% HNK. Ascorbic acid was used as a reference drug (ascorbic acid IC_50_ 3.2 ± 0.2 µg/mL; HNK IC_50_ 5.5 ± 0.3 µg/mL). 

The antioxidant efficacy of MOE was also confirmed in spinal cord samples after oral administration in SNI mice. Neuropathic mice showed overexpression of Nrf2, and MOE treatments showed the capability to reduce the Nrf2 protein content to CTRL values. This effect was not modified by AM251 (Figure 8B).

### 3.8. Effect of MOE on Spinal Expression of MBP

Notch signaling appears to have an increasing role in spinal mechanisms involved in the neuropathogenesis of pain, and the inhibition of Notch-1 leads to the elevation of mechanical and thermal thresholds, which ameliorate pain-related symptoms [41]. In addition, Notch signaling has been reported as a negative regulator of oligodendrocyte differentiation and myelination [42]. We, thus, investigated the effect of MOE treatment on spinal myelin basic protein (MBP) expression. In agreement with recent findings [43], the SNI peripheral neuropathy model did not induce a significant decrease in spinal myelin content. In fact, the level of MBP in the IPSI spinal cord of untreated mice corresponded to the level of MBP in the contra spinal cord. Interestingly, we observed a strong increase in MBP expression in the ipsilateral side in MOE-treated animals (Figure 9). The co-administration of AM251 showed a trend toward the reduction of MBP content that did not reach statistical significance. 

## 4. Discussion

Nowadays, available treatments for neuropathic pain are effective in less than 50 percent of patients because they have numerous side effects that limit their prolonged use [1,44]. The World Health Organization (WHO) reported that more and more people support the use of medicinal plant therapy for neuropathic pain [45]. Based on the high compliance of patients towards herbal medicines and the promising efficacy shown by natural antioxidants in the management of neuropathic pain, in the present work, we investigated the capability of an HNK-rich ethanolic extract of *Magnolia officinalis* (MOE), at antioxidant doses, to modulate pain perception after oral administration in mice. The role of the main constituent HNK in the mechanism of action was also assessed.

*Magnolia officinalis* is largely used in traditional Chinese and Eastern medicine and some of its pharmacological properties, such as antioxidant, anti-inflammatory, antibiotic, and antispasmodic, are also exploited in the clinical practice in the treatment of gastrointestinal disorders, anxiety, depression, and allergies [46]. Neolignans, particularly magnolol, and HNK, represent the major constituents responsible for the biological and pharmacological properties of *Magnolia officinalis* extracts [17,46], mainly related to their antioxidant effects. In the present study, we investigated the analgesic activity of MOE in conditions of acute pain as well as its antiallodynic effect in the SNI model, a condition of trauma-induced NP. Acute oral administration of MOE induced thermal antinociception in naïve mice. We also present evidence on the attenuation of mechanical allodynia on trauma-induced NP by the extract. It has been reported that chronic administration of plant-derived extracts is more effective than the single administration [32,47] and, in future studies, it will be worthwhile to investigate the efficacy of MOE after repeated treatment. MOE pain-relieving activity was devoid of any alteration in the animals’ gross behavior, and the extract did not produce any locomotor impairment.

Increased extracellular glutamate levels following painful stimuli lead to the activation of numerous intracellular pathways, including free radicals’ formation (i.e., ROS), which increase, resulting in oxidative stress. This condition activates cyclooxygenase (COX) enzymes and enhances the production of prostaglandins through the activation of transcription factors (i.e., nuclear factor kappa (NF-kB)), and MAPK [4,6]. MAPKs represent one of the most important signal transduction pathways associated with the onset and development of NP [48]. Indeed, spinal activation of MAPKs has been detected in neurons and glial cells following nerve injury [49]. Consistently, increased phosphorylation of ERK1/2, p38, and JNK1 was observed in the spinal cord dorsal horn of SNI mice. MOE oral administration drastically reduced p-p38 and p-JNK1 levels with no effect on pERK1/2. In NP conditions p38 is largely involved in the modulation of microglia activity [50], while JNK is mainly expressed in astrocyte cells [51]. Thus, a prominent modulation of glia activation by the extract can be hypothesized. 

To further elucidate the mechanism of MOE, the effects on the main proinflammatory mediators were investigated. Increased iNOS expression has been observed in mice with neuropathy [52] and elevated levels of microglial iNOS have been reported in proinflammatory conditions [53,54]. Another important mediator is NF-kB, commonly considered the driver of microglia activation. Inhibitors of this pathway represent promising candidates for neuropathic pain attenuation [55]. A reversal of NF-kB, iNOS, and IL-1ß overexpression in spinal cord samples from SNI following MOE administration was observed, along with an overexpression of the anti-inflammatory cytokine IL-10 [56], further indicating the capability of MOE to attenuate spinal neuroinflammation. To confirm this hypothesis and to assess the role of the main constituent HNK in the MOE activity, the effect of the neolignan on LPS-stimulated microglial BV2 cells was evaluated. LPS-stimulated BV2 cells were used as a validated in vitro model for reproducing the effect of microglia activation in animals, and they have been used to investigate the role of neuroinflammation in models of neuropathic pain [57]. HNK restored the cell viability and homeostatic morphology in LPS-stimulated cells, in addition to reducing the number of cells in the pro-inflammatory state. These results are supported by previous studies on BV2 cells stimulated with INFγ + LPS and on primary cultures of microglia and astrocytes stimulated by LPS in which HNK displayed anti-inflammatory activity by reducing pro-inflammatory mediators (iNOS, IL-6, IL-1β, and TNF-α) [22,58]. In another study, HNK abrogated the production of LPS-stimulated TNF-α and inhibited the activity of NF-κB, COX-2, and peroxidation of lipids in mouse monocytes [59]. Moreover, the antioxidant efficacy of MOE and HNK, administered at the concentration present in the active dose of the extract, were comparable. Even if MOE contains numerous constituents with potential biological activity, these results further support the hypothesis of HNK as the main active biomolecule.

Several studies have shown that neolignans present in the bark of *Magnolia officinalis* interact with cannabinoid (CB) receptors and HNK has shown full agonistic activity at CB1 receptors [25]. The presence of CB1 in the central nervous system (CNS) plays a crucial role in controlling normal synaptic activity, which, if altered, can underlie the onset of numerous pathologies, including neurodegenerative diseases and behavioral disorders [60] An important role in neuropathic pain perception is attributed to CB1, as in vivo studies showed that reducing the activity of these receptors results in increased pain sensation [61], and today, the endocannabinoid system represents one of the main targets for the modulation of pathways involved in neuropathic pain [62]. 

CB1 involvement in the pain-relieving effect of MOE was evaluated using treatment with AM251, a known antagonist of this receptor class. Possible modulation of the endocannabinoid system in NP conditions by the extract was indicated by the prevention of the anti-allodynic effect of MOE in the presence of the CB1 antagonist. Concerning the molecular mechanisms involved, AM251 treatment selectively prevented p-p38, iNOS, NF-kB, and IL-1ß increase, indicating a prevalent involvement of the CB1-mediated endocannabinoid system in the MOE-induced attenuation of pro-inflammatory events.

HNK was found to produce antitumor activity through the modulation of the Notch signaling pathway. The inhibition of cell proliferation, colony formation, and colon cancer markers by inhibiting the γ-secretase complex and the Notch signaling pathway was observed [63]. Furthermore, treatment with HNK potently inhibited melanoma cells by reducing the expression of cleaved Notch and downstream proteins [64]. Several studies showed that the Notch signaling pathway is involved in the promotion of NP in the SNI model [65,66]. Treatment with Notch inhibitors attenuates pain hypersensitivity in trauma-induced [67] and diabetic [68] neuropathies. Notch signaling occurs between two adjacent cells, a signal-sending cell that presents the ligand (Delta-like 1, 3, and 4 and Jagged 1 and 2) and a signal-receiving cell that expresses the Notch receptor [69]. By investigating the role of the Notch signaling pathway in the mechanism of MOE, in spinal cord samples from SNI mice we observed an increased expression of Jagged1, the Notch ligand, and of the membrane-anchored Notch extracellular truncation (NEXT) fragment which is generated by Notch cleavage following Notch receptor activation. In MOE-treated mice, we observed a reduction in the overexpression of the above-mentioned proteins. The role of HNK in this signaling cascade was investigated in LPS-stimulated BV2 cells that have been reported to express Jagged1 [70,71]. Both MOE and HNK treatments drastically reduced the Jagged1 overexpression, further confirming the involvement of the Notch signaling pathway in the MOE effect. Furthermore, the present findings let us hypothesize that the modulation of the Notch pathway is related to the content in HNK.

## 5. Conclusions

In conclusion, the present findings illustrated the efficacy of an HNK-rich MOE in attenuating NP symptoms without inducing any locomotor impairment. This activity was produced at antioxidant doses and appears to be related to the capability of the extract and of its main constituent HNK to reduce neuroinflammation through a CB1-mediated mechanism. Thus, MOE and HNK, by modulating oxidative and proinflammatory responses, might represent interesting candidates for NP management.

## Figures and Tables

**Figure 1 antioxidants-12-01518-f001:**
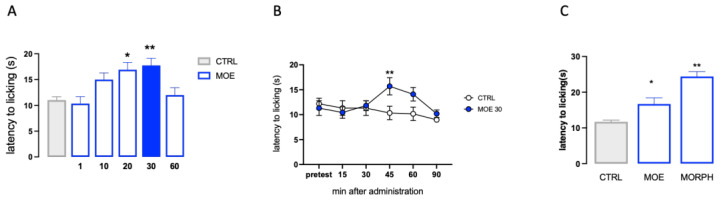
Analgesic activity of MOE in a condition of acute thermal pain. (**A**) Dose-response curve for the analgesic effect of MOE (1–60 mg/kg p.o.) in the hot plate test, the blue bar indicated the dose used. (**B**) Time-course curve for the analgesic effect of MOE (30 mg/kg p.o.). (**C**) Comparison of MOE activity vs. morphine (MORPH; 7 mg/kg s.c.). CTRL: control untreated group. * *p* < 0.05, ** *p* < 0.01 vs. CTRL.

**Figure 2 antioxidants-12-01518-f002:**
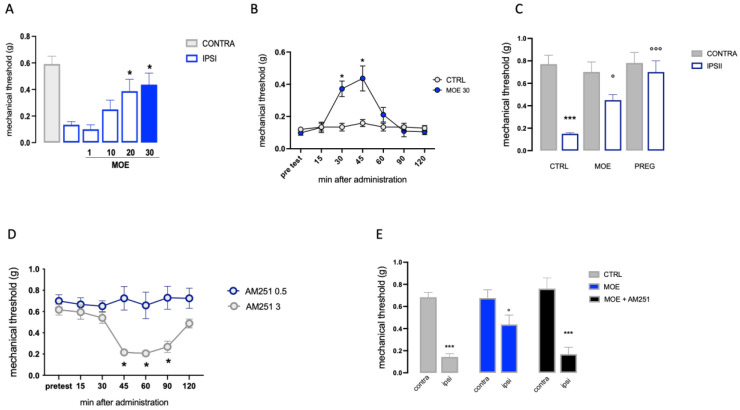
The antiallodynic activity of MOE in SNI mice. (**A**) Dose-response curve for the attenuation of mechanical hypersensitivity in SNI mice by MOE (1–30 mg/kg p.o.), the blue bar indicated the dose used. (**B**) Time-course evaluation of the increase in the mechanical threshold by MOE 30 mg/kg. (**C**) Comparison of the MOE activity with pregabalin (PREG) used as a reference drug. (**D**) Determination of the non-hyperalgesic dose of AM251 in the uninjured contralateral side. (**E**) Prevention of MOE antiallodynic effect by AM251 (0.5 mg/kg). CONTRA: contralateral side; IPSI: ipsilateral side; CTR: control group. * *p* < 0.05, *** *p* < 0.001 vs. CTRL; ° *p* < 0.05, °°° *p* < 0.001 vs. IPSI.

**Figure 3 antioxidants-12-01518-f003:**
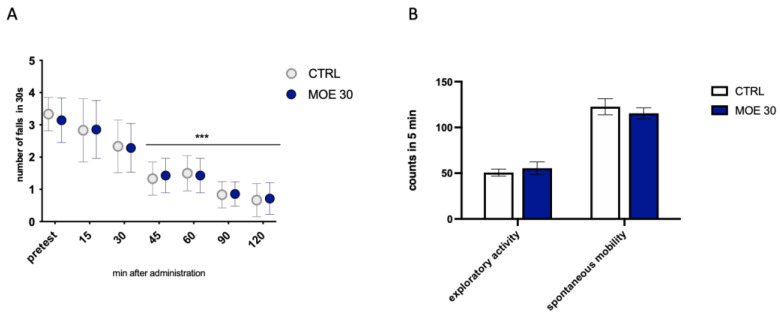
Effect of MOE on locomotor behavior. (**A**) Lack of effect of MOE (30 mg/kg) on motor coordination evaluated by the rotarod test. The test was performed before and 15, 30, 45, 60, 90, and 120 min after oral administration in naïve mice. (**B**) Lack of impairment of spontaneous mobility and exploratory activity by MOE (30 mg/kg) in the hole board test. The test was performed 45 min after administration. *** *p* < 0.001 vs. CTRL.

**Figure 4 antioxidants-12-01518-f004:**
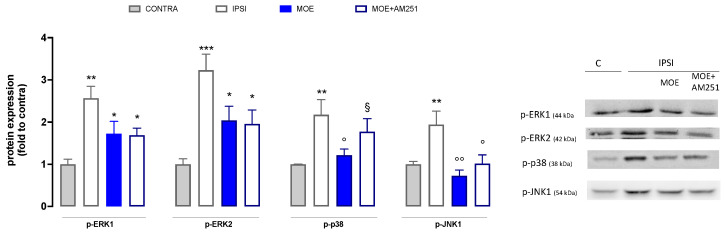
Effect of MOE on MAPK phosphorylation in spinal cord tissue from SNI mice. Protein levels were evaluated by western blot experiments. Representative blots are reported. CONTRA: contralateral side; IPSI: ipsilateral side. * *p* < 0.05, ** *p* < 0.01, *** *p* < 0.001 vs. CONTRA; ° *p* < 0.05, °° *p* < 0.01 vs. IPSI; § *p* < 0.05 vs. IPSI + MOE.

**Figure 5 antioxidants-12-01518-f005:**
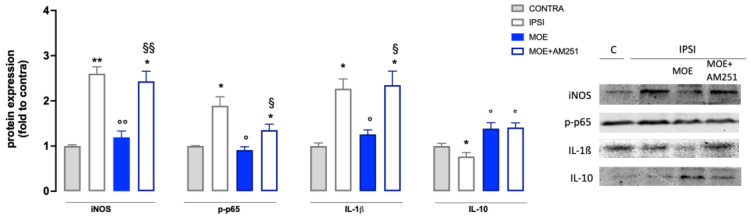
Effect of MOE on the expression of proinflammatory and anti-inflammatory mediators in the spinal cord of SNI mice. Protein levels were evaluated by western blot experiments. Representative blots are reported. CONTRA: contralateral side; IPSI: ipsilateral side. * *p* < 0.05, ** *p* < 0.01 vs. CONTRA; ° *p* < 0.05, °° *p* < 0.01 vs. IPSI; § *p* < 0.05 §§ *p* < 0.01vs. IPSI + MOE.

**Figure 6 antioxidants-12-01518-f006:**
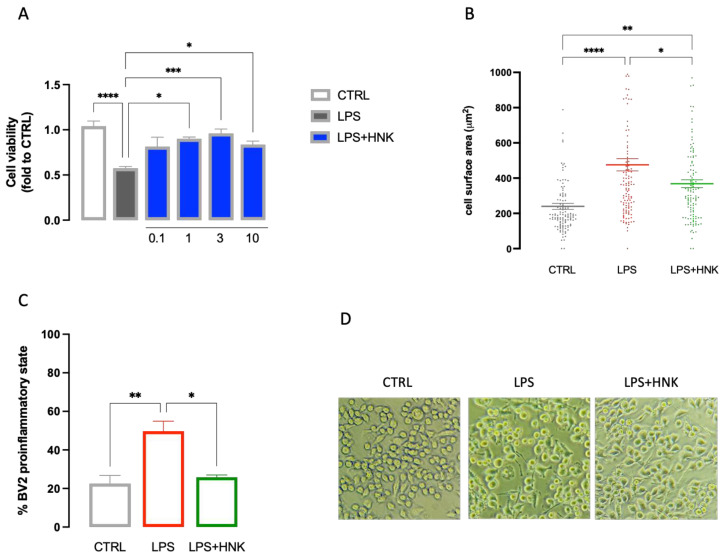
Anti-inflammatory effect of HNK on LPS-stimulated BV2 cells. (**A**) Dose-response curve for the prevention by HNK (0.1–10 µg/mL) of cell viability reduction induced by LPS stimulation (250 ng for 24 h). (**B**) HNK (3 µg/mL) reduced the increase in the cell surface induced by LPS stimulation. (**C**) HNK (3 µg/mL) reduced the number of BV2 cells in the proinflammatory state after LPS stimulation. (**D**) Scale bar 100 µM. Representative images of LPS-stimulated BV2 cells. * *p* < 0.05, ** *p* < 0.01, *** *p* < 0.001, **** *p* < 0.0001.

**Figure 7 antioxidants-12-01518-f007:**
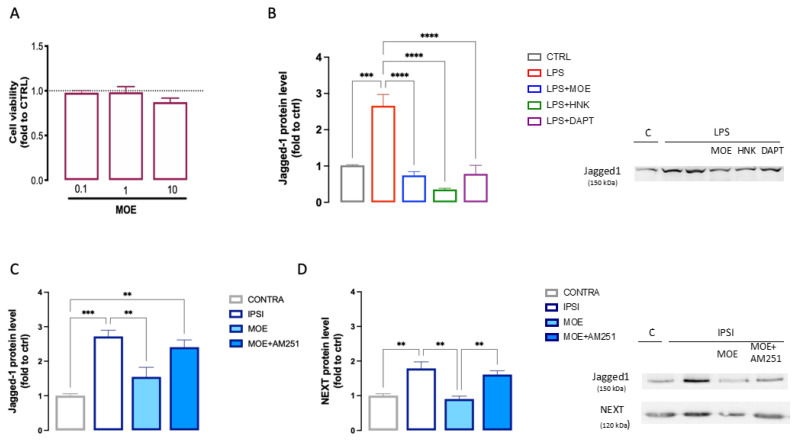
Attenuation of MOE and HNK of the Notch signaling pathway. (**A**) Lack of impairment of cell viability by MOE (0.1–10 µg/mL) evaluated by the CCK-8 test in BV2 cells. (**B**) Reduction by MOE (10 µg/mL), HNK (3 µg/mL), and DAPT (3 µM) of the overexpression of Jagged1 induced by LPS stimulation in BV2 cells. Prevention of Jagged1 (**C**) and NEXT (**D**) overexpression by MOE treatment in spinal cord samples of SNI mice. Lack of effect by AM251 administration. Representative blots are reported. ** *p* < 0.01, *** *p* < 0.001, **** *p* < 0.0001.

**Figure 8 antioxidants-12-01518-f008:**
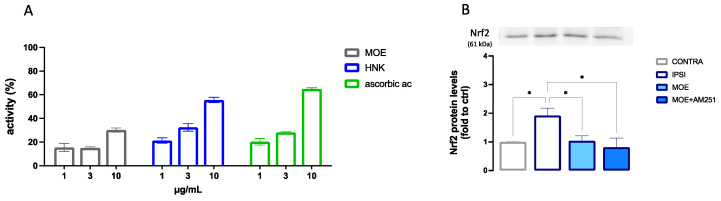
Antioxidant activity of HNK and MOE. (**A**) Anti-radical scavenger activity of MOE and HNK (1–10 µg/µL) in the DPPH test. Ascorbic acid was used as a reference drug. (**B**) Reduction of the overexpression of Nfr2 by MOE in spinal cord samples from SNI mice. * *p* < 0.05.

**Figure 9 antioxidants-12-01518-f009:**
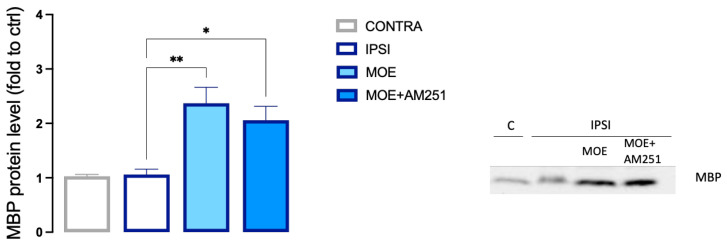
MOE-induced increase of MBP content in the spinal cord of SNI mice. Lack of effect by AM251 treatment. MBP values were obtained by Western blot analysis. Representative blots are reported. * *p* < 0.05, ** *p* < 0.01.

## Data Availability

The data presented in this study are available on request from the corresponding author.

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
