# Peer review of "Honokiol-Rich Magnolia officinalis Bark Extract Attenuates Trauma-Induced Neuropathic Pain"

_antioxidants, 2023, doi:10.3390/antiox12081518_

Round 1

Reviewer 1 Report

In this paper, Borgonetti and Galeotti, investigate the anti-nociceptive properties of the extract of Magnolia Officinalis (MOE), an herbal compound widely used in Eastern medicine for its antioxidant and anti-inflammatory activity. They study the molecular mechanism involved both in vitro using a microglial cell line and in vivo using a well-known model of neuropathic pain, the SNI model. Furthermore, they demonstrate that the anti-nociceptive activity involves the cannabinoid system.

Even if I found the basal hypothesis very interesting,I think the paper needs a substantial revision before been suitable for publication.

1.       First of all it look that part of the results are missing (line 448).

2.       The blots have to be re-done:

·       I completely disagree with cutting single bands to put the sample in the right order

·       For each blot the ponceux has to be shown or the blot for a housekeeping gene has to be performed and shown

·       Are the antibodies for different signaling proteins put sequentially on the same membrane? Or are different membranes? Please specify.

·       The original gels related to Fig 9 are missing.

·       It looks something went wrong with cutting and past for the blot for NEXT in fig 7: the bands do not correspond to the ones in the original blot.

3.       The results of the hot plate experiment performed in untreated mice (Fig 1a) show that MOE alters the basal sensitivity of nociception, not acute pain. To check the effect on acute pain other test are needed, like capsaicin test

4.       SNI injury is known to alter more the cold than the warm sensitivity; a cold plate test is more appropriate to test thermal sensitivity after SNI operation

5.       In Fig 2D the effect of AM251 administrated alone is shown as latency of liking (s), on line 289 is described as effect on pain threshold, previously measured in g. Which test has been used?

6.       Please discuss the very rapid drop in the anti-nociceptive effect of MOE at 60 min?

7.       In Fig 4 the sample CONTRA is taken from sham-operated mice, SNI mice, or SNI treated with MOE? MOE alters basal nociception so is expected that it alters signaling protein also in the contralateral side since is given systemically

8.       Line 368: here looks like the cells are first treated with LPS and then with the different compounds, in the methods section looks the opposite; what is correct?

9.       Line 243: the number of independent experiments changes for each test

10.   Why only male mice are used? The authors hypothesize that microglial cells are involved in the anti-nociceptive effect of MOE. It is known that microglia cells show gender-dependent behavior in neuropathic conditions. Do they expect the same effect on female mice? They should at least discuss this issue.

Reviewer 2 Report

In order to improve the manuscript, several changes must be included in the final version:

INTRODUCCIÓN

The IASP defines chronic pain as pain that persists for more than 3 months. The origin of chronic pain can be induced by cancer, post-surgical pain, neuropathic pain, nociplastic pain, visceral pain, musculoskeletal pain, etc. Authors should review the dichotomous classification of the first sentence of the introduction.

In addition to including references that are reviews, the authors could also include previous studies where antioxidant substances such as polyphenolic substances have been used in experimental models of neuropathic pain-inducing peripheral nerve injury or neuropathic pain-inducing spinal cord injury. Likewise, in the scientific bibliography there is an article where Magnolia officinalis is used for the treatment of neuropathic pain caused by chemotherapeutics (Plants (Basel). 2023 Jun 12;12(12):2283.). It is very interesting for the readers that the authors collect these previous articles on the use of polyphenols and other antioxidant molecules in the treatment of peripheral neuropathic pain of various origins. Please include this information in the final version of the manuscript.

MATERIALS AND METHODS

The authors must indicate the Ethics Committee that approved the animal experimentation procedure used in the study, indicating the date of acceptance. This information is essential and must be included in the final version of the manuscript.

RESULTS

The authors indicate “The efficacy of MOE on attenuating the nociceptive behavior was evaluated in conditions of both acute and chronic pain”. Bearing in mind that all the functional evaluations of pain have been carried out 7 days after the sciatic nerve injury, the authors should define what they consider to be the acute and chronic phase of pain. In other words, what is acute pain condition and chronic pain condition for the authors? Pain responses that appear a week after causing a nerve injury are considered acute pain or chronic pain? Please, include this information in the final version of the manuscript.

DISCUSSION

The chemical analysis of the extracts of Magnolia officinalis shows that they contain various products such as: (S)-4-keto-magnoflorine, (R)-3,4-dehydromagnocurarine, magnoloside A, magnoloside B, magnoloside F, syringin, (- )-α-pinene and β-pinene, crassifolioside, magnolol, obvatol, among others, in addition to those described by the authors, which should be the most abundant. Which of these compounds, the majority and the minority, have anti-oxidant and analgesic effects? This information is relevant and should be discussed in the final version of the manuscript.

Oral administration of Magnolia officinalis extracts generates a series of metabolites observed in urine such as: sinapic acid-4-O-sulfate, sinapic acid-4-O- - glucuronide, sinapic acid, 3-[20,6-dihydroxy- 5 0-(2-propenyl)[1,10-biphenyl]-3-yl]-(E)-2-propenoic acid, and the unchanged form magnolol (J Pharm Pharmacol. 2003 Nov;55(11):1583-91). This means that at the hepatic level the extract and its components are modified, generating these metabolites that are purified via the kidneys. Do some of these metabolites from Magnolia officinalis extracts have analgesic effects? The analgesic effect of the extract can be attributable only to the original components of the extract or also to the metabolites that are generated? The authors could discuss this point in the final version of the manuscript.

The authors administer the extract 45 minutes before assessing pain, and the results they observe is a maximum peak of analgesic effect, which disappears over time. What potential clinical utility can a treatment of neuropathic pain have with a product like this extract whose analgesic effects are extremely short? Considering the experience of the authors in the fields of pharmacology and toxicology, what should be done to prolong the analgesic effect of these extracts? These two points could also be discussed in the final version of the manuscript.

There is previous evidence that some of the compounds in the Magnolia officinalis extract have toxic effects. This point could also be discussed in the context of the potential applicability of this extract in patients with peripheral neuropathic pain.

There are previous studies that show analgesic effects when administering Honokiol, the main component of the Magnolia officinalis extract. The authors may wish to discuss these previous studies in the context of neuropathic pain. Likewise, it could also be interesting to discuss whether it is more effective to directly administer Honokiol or better the Magnolia officinalis extract in the context of relieving peripheral neuropathic pain.

Seven days after inducing peripheral nerve injury, the authors administered MOE orally 45 minutes before performing functional pain tests. After completing these tests, they sacrifice the animals, extract the spinal cord and assess molecular changes. The results show variations in various pro- and anti-inflammatory parameters. The authors should discuss how MOE exerts this effect on the observed molecular changes, keeping in mind that it must be absorbed intestinally, make it to enter the spinal cord with an intact blood-spinal barrier, and act on cellular elements of the spinal cord to cause changes in the expression of the analyzed molecular parameters. All this information should be included in the final version of the manuscript.

The authors perform in vitro studies with a microglia cell line to see the effects of the extract. It is surprising that, having used spinal cord from nerve-injured animals for molecular studies, they did not perform histological studies with IBA1 labeling, which allows visualization of microglia cells from the spinal cord of nerve-injured animals. Despite this, the authors could discuss whether previous studies with a lesion identical to the one used in the manuscript have found microglia changes in the spinal cord. And discuss whether the changes described by these authors are like those observed by the authors in microglia cultures, and therefore infer that what is observed in vitro is like what is observed in vivo. It should be noted that there are two previous articles that have studied the effect of the Magnolia officinalis extract or main components on BV2 cell cultures. Please include this information in the final version of the manuscript.

Round 2

Reviewer 1 Report

Although I believe that the manuscript has improved a lot, believe it is not yet ready for publication.

First of all, I am not completely satisfied with the answers to my previous doubts, particularly again with the Western blots:

-          Supplementary Fig 8-9: if the 2 blots came from the same membrane (only 1 ponceaux is shown), please include the IPSI-MOE+AM251 sample also for Nrf2 blot.

-          Supplementary Fig 4: If I understand correctly one membrane was used for pErk and one for pP38 and pJNK. The examples do not support what is described in the text-.

As for pP38, it seems that AM251 has a minor effect on the effect of MOE. It seems that MOE increases pErk1. There seems to be no effect of MOE (or MOE + AM251) on the expression of p-JNK.

Molecular weights are incorrectly stated: I mean higher molecular weights run slower in the gel so they are higher than smaller molecular weights.

-          Supplementary Fig 5: it looks like the shown bands for IL-1ß do not correspond to the shown original gel.

-          Supplementary Fig 7: blot for Jagged in BV2 cells: it is again cut. Since it stands alone there is no reason to cut it, also from the ponceaux it looks like the LPS sample is overloaded compared to the others

-          Fig 9: Are MOE and MOE+AM251 treatments statistically different?

Since all behavior data are shown as mechanistic hypersensitivity it would be useful to show that AM251 does not alter baseline mechanical sensitivity in naive mice rather than thermal or otherwise both. The same is true for the effect of MOE in naive mice.

In the test or fig legend there is no reference that results comparable with thermal stimulation, were obtained by performing a dose-response curve for AM251 against mechanical stimulation.

Since MOE alters the basal nociceptive threshold, it must be shown that in mice treated with MOE under sham conditions or SNI operation, there is no alteration of signaling molecules in the contralateral part

If the reason for using only male mice lies in planning the study to analyze the effect on microglia, this must be explicitly stated already in the abstract

In general, there is a lack of evidence that THE analgesic effect of MOE is mediated by microglia. The authors should mitigate the statement or strengthen the link such as showing co-staining of modulation of signaling molecules with Iba1 marker of microglia on spinal cord tissues. Or at least strengthen the link between in vivo and in vitro experiments such as analyzing the modulation of signaling pathways involved in vivo in BV2 cells treated with MOE.

Minor points:

Please better specify what you mean by BV2 activated state (line 390): how is it evaluated? through morphological analysis or also other parameters?

Line 387: please use only one unit of measurement for HNK µg/ml or µM. It needs to be standardized

Line 422 attenuate statement: aM251 partially reverses the effect of MOE on Jagged1 expression

The description of the following Ab is missing in material and methods: Nrf2, MBP, ßAct, NEXT

Reviewer 2 Report

The authors have made an effort to respond to all of the reviewer's suggestions and comments, including many of them in the final version of the manuscript.

Author Response

We thank the reviewer for his/her positive comment on the revised version of the manuscript